# CAN DEEP REINFORCEMENT LEARNING SOLVE ERDOS-SELFRIDGE-SPENCER GAMES?

## ABSTRACT

Deep reinforcement learning has achieved many recent successes, but our understanding of its strengths and limitations is hampered by the lack of rich environments in which we can fully characterize optimal behavior, and correspondingly diagnose individual actions against such a characterization. Here we consider a family of combinatorial games, arising from work of Erdos, Selfridge, and Spencer, and we propose their use as environments for evaluating and comparing different approaches to reinforcement learning. These games have a number of appealing features: they are challenging for current learning approaches, but they form (i) a low-dimensional, simply parametrized environment where (ii) there is a linear closed form solution for optimal behavior from any state, and (iii) the difficulty of the game can be tuned by changing environment parameters in an interpretable way. We use these Erdos-Selfridge-Spencer games not only to compare different algorithms, but also to compare approaches based on supervised and reinforcement learning, to analyze the power of multi-agent approaches in improving performance, and to evaluate generalization to environments outside the training set.

## 1 INTRODUCTION

Deep reinforcement learning has seen many remarkable successes over the past few years (Mnih et al., 2015) (Silver et al., 2017). But developing learning algorithms that are robust across tasks and policy representations remains a challenge. Standard benchmarks like MuJoCo and Atari provide rich settings for experimentation, but the specifics of the underlying environments differ from each other in many different ways, and hence determining the principles underlying any particular form of sub-optimal behavior is difficult. Optimal behavior in these environments is generally complex and not fully characterized, so algorithmic success is generally associated with high scores, making it hard to analyze where errors are occurring in any sort of fine-grained sense.

An ideal setting for studying the strengths and limitations of reinforcement learning algorithms would be (i) a simply parametrized family of environments where (ii) optimal behavior can be completely characterized, (iii) the inherent difficulty of computing optimal behavior is tightly controlled by the underlying parameters, and (iv) at least some portions of the parameter space produce environments that are hard for current algorithms. To produce such a family of environments, we look in a novel direction – to a set of two-player combinatorial games with their roots in work of Erdos and Selfridge (Erdos & Selfridge, 1973), and placed on a general footing by Spencer (1994). Roughly speaking, these *Erdos-Selfridge-Spencer (ESS) games* are games in which two players take turns selecting objects from some combinatorial structure, with the feature that optimal strategies can be defined by potential functions derived from conditional expectations over random future play.

These ESS games thus provide an opportunity to capture the general desiderata noted above, with a clean characterization of optimal behavior and a set of instances that range from easy to very hard as we sweep over a simple set of tunable parameters. We focus in particular on one of the best-known games in this genre, *Spencer's attacker-defender game* (also known as the "tenure game"; Spencer, 1994), in which — roughly speaking — an *attacker* advances a set of pieces up the levels of a board, while a *defender* destroys subsets of these pieces to try prevent any of them from reaching the final level (Figure 1). An instance of the game can be parametrized by two key quantities. The first is the number of levels $K$, which determines both the size of the state space and the approximate length of the game; the latter is directly related to the sparsity of win/loss signals as rewards. The second

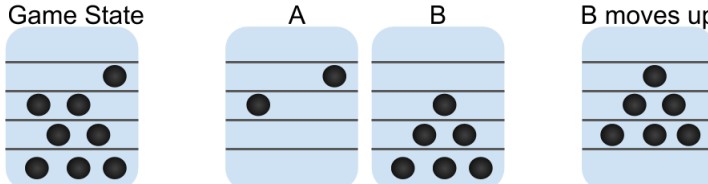

Figure 1: One turn in an ESS Attacker-Defender game. The attacker proposes a partition $A, B$ of the current game state, and the defender chooses one set to destroy (in this case $A$). Pieces in the remaining set ($B$) then move up a level to form the next game state.

quantity is a *potential function* $\phi$, whose magnitude characterizes whether the instance favors the defender or attacker, and how much "margin of error" there is in optimal play.

The environment therefore allows us to study learning by the defender, or by the attacker, or in a multi-agent formulation where the defender and attacker are learning concurrently. Because we have a move-by-move characterization of optimal play, we can go beyond simple measures of reward based purely on win/loss outcomes and use supervised learning techniques to pinpoint the exact location of the errors in a trajectory of play. In the process, we are able to develop insights about the robustness of solutions to changes in the environment. These types of analyses have been long-standing goals, but they have generally been approached much more abstractly, given the difficulty in characterizing step-by-step optimally in non-trivial environments such as this one.

The main contributions of this work are thus the following:

1. The development of these combinatorial games as environments for studying the behavior of reinforcement learning algorithms, with sensitive control over the difficulty of individual instances using a small set of natural parameters.

2. A comparison of the performance of an agent trained using deep RL to the performance of an agent trained using supervised learning on move-by-move decisions. Exploiting the fact that we can characterize optimal play at the level of individual moves, we find an intriguing phenomenon: while the supervised learning agent is, not surprisingly, more accurate on individual move decisions than the deep RL agent, the deep RL agent is better at playing the game! We further interpret this result by studying *fatal mistakes*.

3. An investigation of the way in which the success of one of the two players (defender or attacker) in training turns out to depend crucially on the algorithm being used to implement the other player. We explore properties of this other player's algorithm, and also properties of mulitagent learning, that lead to more robust policies with better generalization.

This is a largely empirical paper, building on a theoretically grounded environment derived from a combinatorial game. We present learning and generalization experiments for a variety of commonly used model architectures and learning algorithms. We aim to show that despite the simple structure of the game, it provides both significant challenges for standard reinforcement learning approaches and a number of tools for precisely understanding those challenges.

## 2 ERDOS-SELFRIDGE-SPENCER ATTACKER DEFENDER GAME

We first introduce the family of Attacker-Defender Games (Spencer, 1994), a set of games with two properties that yield a particularly attractive testbed for deep reinforcement learning: the ability to continuously vary the difficulty of the environment through two parameters, and the existence of a closed form solution that is expressible as a *linear model*.

An Attacker-Defender game involves two players: an attacker who moves pieces, and a defender who destroys pieces. An instance of the game has a set of *levels* numbered from 0 to $K$, and $N$ pieces that are initialized across these levels. The attacker's goal is to get at least one of their pieces to level $K$, and the defender's is to destroy all $N$ pieces before this can happen. In each turn, the attacker proposes a partition $A, B$ of the pieces still in play. The defender then chooses one of the sets to

destroy and remove from play. All pieces in the other set are moved up a level. The game ends when either one or more pieces reach level $K$, or when all pieces are destroyed. Figure 1 shows one turn of play.

With this setup, varying the number of levels $K$ or the number of pieces $N$ changes the difficulty for the attacker or the defender. One of the most striking aspects of the Attacker-Defender game is that it is possible to make this tradeoff precise, and en route to doing so, also identify a *linear optimal policy*.

We start with a simple special case — rather than initializing the board with pieces placed arbitrarily, we require the pieces to all start at level 0. In this special case, we can directly think of the game's difficulty in terms of the number of levels $K$ and the number of pieces $N$.

**Theorem 1.** *Consider an instance of the Attacker-Defender game with $K$ levels and $N$ pieces, with all $N$ pieces starting at level 0. Then if $N < 2^K$, the defender can* always *win.*

There is a simple proof of this fact: the defender simply always destroys the larger one of the sets $A$ or $B$. In this way, the number of pieces is reduced by at least a factor of two in each step; since a piece must travel $K$ steps in order to reach level $K$, and $N < 2^K$, no piece will reach level $K$.

When we move to the more general case in which the board is initialized at the start of the game with pieces placed at arbitrary levels, it will be less immediately clear how to define the "larger" one of the sets $A$ or $B$. We therefore give a second proof of Theorem 1 that will be useful in these more general settings. This second proof (Spencer, 1994) uses Erdos's probabilistic method and proceeds as follows: for any attacker strategy, assume the defender plays randomly. Let $T$ be a random variable for the number of pieces that reach level $K$. Then $T = \sum T_i$ where $T_i$ is the indicator that piece $i$ reaches level $K$.

But then $E[T] = \sum E[T_i] = \sum_i 2^{-K}$: as the defender is playing randomly, any piece has probability $1/2$ of advancing a level and $1/2$ of being destroyed. As all the pieces start at level 0, they must advance $K$ levels to reach the top, which happens with probability $2^{-K}$. But now, by choice of $N$, we have that $\sum_i 2^{-K} = N2^{-K} < 1$. Since $T$ is an integer random variable, $E[T] < 1$ implies that the distribution of $T$ has nonzero mass at 0 - in other words there is some set of choices for the defender that guarantees destroying all pieces. This means that the attacker does not have a strategy that wins with probability 1 against random play by the defender; since the game has the property that one player or the other must be able to force a win, it follows that the defender can force a win. □

Now consider the general form of the game, in which the initial configuration can have pieces at arbitrary levels. Thus, at any point in time, the state of the game can be described by a $K$-dimensional vector $S = (n_0, n_1, ..., n_K)$, with $n_i$ the number of pieces at level $i$.

Extending the argument used in the second proof above, we note that a piece at level $l$ has a $2^{-(K-l)}$ chance of survival under random play. This motivates the following *potential function* on states:

**Definition 1.** Potential Function*: Given a game state $S = (n_0, n_1, ..., n_K)$, we define the* potential *of the state as $\phi(S) = \sum_{i=0}^{K} n_i 2^{-(K-i)}$.*

Note that this is a *linear* function on the input state, expressible as $\phi(S) = w^T \cdot S$ for $w$ a vector with $w_l = 2^{-(K-l)}$. We can now state the following generalization of Theorem 1.

**Theorem 2.** *Consider an instance of the Attacker-Defender game that has $K$ levels and $N$ pieces, with pieces placed anywhere on the board, and let the initial state be $S_0$. Then*

   *(a) If $\phi(S_0) < 1$, the defender can always win*

   *(b) If $\phi(S_0) \geq 1$, the attacker can always win.*

One way to prove part (a) of this theorem is by directly extending the proof of Theorem 1, with $E[T] = \sum E[T_i] = \sum_i 2^{-(K-l_i)}$ where $l_i$ is the level of piece $i$. After noting that $\sum_i 2^{-(K-l_i)} = \phi(S_0) < 1$ by our definition of the potential function and choice of $S_0$, we finish off as in Theorem 1.

This definition of the potential function gives a natural, concrete strategy for the defender: the defender simply destroys whichever of $A$ or $B$ has higher potential. We claim that if $\phi(S_0) < 1$, then this strategy guarantees that any subsequent state $S$ will also have $\phi(S) < 1$. Indeed, suppose

(renaming the sets if necessary) that $A$ has a potential at least as high as $B$'s, and that $A$ is the set destroyed by the defender. Since $\phi(B) \leq \phi(A)$ and $\phi(A) + \phi(B) = \phi(S) < 1$, the next state has potential $2\phi(B)$ (double the potential of $B$ as all pieces move up a level) which is also less than 1. In order to win, the attacker would need to place a piece on level $K$, which would produce a set of potential at least 1. Since all sets under the defender's strategy have potential strictly less than 1, it follows that no piece ever reaches level $K$.

If $\phi(S_0) \geq 1$, we can devise a similar optimal strategy for the attacker. The attacker picks two sets $A, B$ such that each has potential $\geq 1/2$. The fact that this can be done is shown in Theorem 3, and in Spencer (1994). Then regardless of which of $A, B$ is destroyed, the other, whose pieces all move up a level, doubles its potential, and thus all subsequent states $S$ maintain $\phi(S) \geq 1$, resulting in an eventual win for the attacker.

## 3   RELATED WORK

The Atari benchmark (Mnih et al., 2015) is a well known set of tasks, ranging from easy to solve (Breakout, Pong) to very difficult (Montezuma's Revenge). Duan et al. (2016) proposed a set of continuous environments, implemented in the MuJoCo simulator Todorov et al. (2012). An advantage of physics based environments is that they can be varied continuously by changing physics parameters (Rajeswaran et al., 2016), or by randomizing rendering (Tobin et al., 2017). Deepmind Lab (Beattie et al., 2016) is a set of 3D navigation based environments. OpenAI Gym (Brockman et al., 2016) contains both the Atari and MuJoCo benchmarks, as well as classic control environments like Cartpole (Stephenson, 1909) and algorithmic tasks like copying an input sequence. The difficulty of algorithmic tasks can be easily increased by increasing the length of the input. Our proposed benchmark merges properties of both the algorithmic tasks and physics-based tasks, letting us increase difficulty by discrete changes in length or continuous changes in potential.

## 4   DEEP REINFORCEMENT LEARNING ON THE ATTACKER-DEFENDER GAME

From Section 2, we see that the Attacker-Defender games are a family of environments with a difficulty knob that can be continuously adjusted through the start state potential $\phi(S_0)$ and the number of levels $K$. In this section, we describe a set of baseline results on Attacker-Defender games that motivate the exploration in the remainder of this paper. We set up the Attacker-Defender environment as follows: the game state is represented by a $K + 1$ dimensional vector for levels 0 to $K$, with coordinate $l$ representing the number of pieces at level $l$. For the defender agent, the input is the concatenation of the partition $A, B$, giving a $2(K + 1)$ dimensional vector. The game start state $S_0$ is initialized randomly from a distribution over start states of a certain potential.

### 4.1   TRAINING A DEFENDER AGENT ON VARYING ENVIRONMENT DIFFICULTIES

We first look at training a defender agent against an attacker that randomly chooses between (mostly) playing optimally, and (occasionally) playing suboptimally, with the *Disjoint Support Strategy*. This strategy unevenly partitions the occupied levels between $A, B$ so that one set has higher potential than the other, with the proportional difference between the two sets being sampled randomly. Note that this strategy gives rise to very different states $A, B$ (uneven potential, disjoint occupied levels) than the optimal strategy, and we find that the model learns a much more generalizable policy when mixing between the two (Section 6).

When testing out reinforcement learning, we have two choices of difficulty parameters. The potential of the start state, $\phi(S_0)$, changes how optimally the defender has to play, with values close to 1 giving much less leeway for mistakes in valuing the two sets. Changing $K$, the number of levels, directly affects the sparsity of the reward, with higher $K$ resulting in longer games and less feedback. Additionally, $K$ also greatly increases the number of possible states and game trajectories (see Theorem 4).

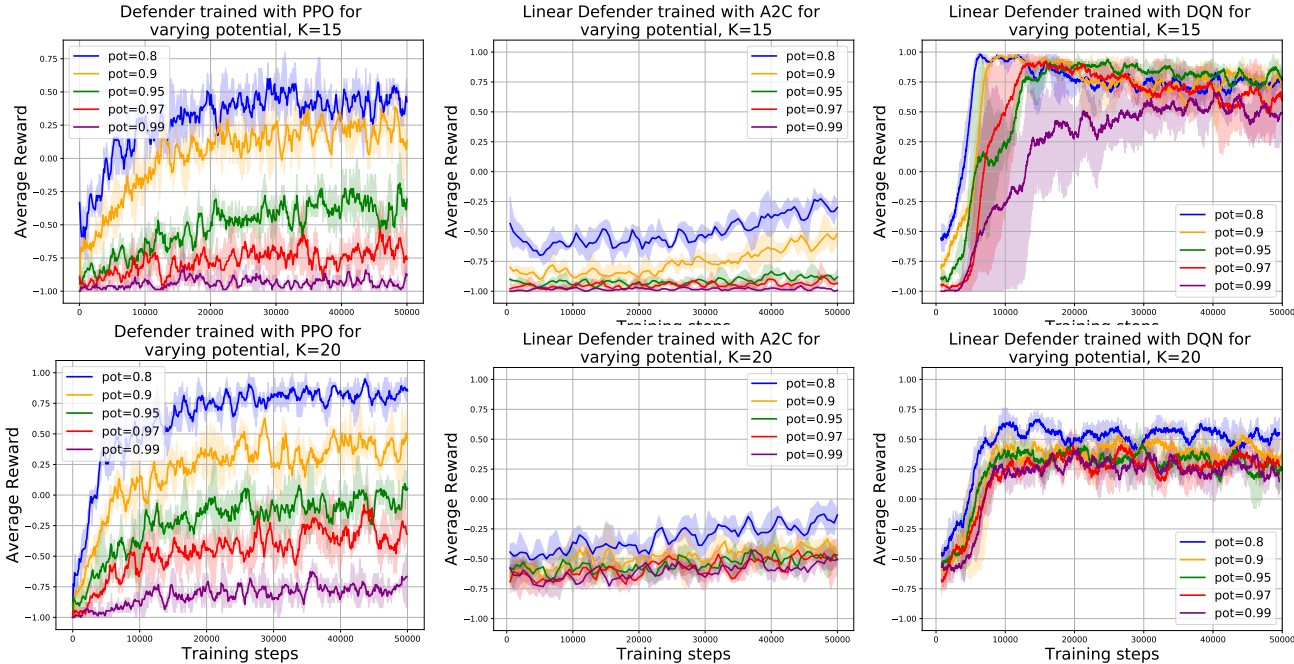

Figure 2: Training a linear network to play as the defender agent with PPO, A2C and DQN. A linear model is theoretically expressive enough to learn the optimal policy for the defender agent. In practice, we see that for many difficulty settings and algorithms, RL struggles to learn the optimal policy and performs more poorly than when using deeper models (compare to Figure 3). An exception to this is DQN which performs relatively well on all difficulty settings.

#### 4.1.1 WARM UP: A LINEAR BASELINE

Recall that the optimal policy can be expressed as a linear network, with the weights given by the potential function, Definition 1. We therefore first try training a linear model for the defender agent. We evaluate Proximal Policy Optimization (PPO) (Schulman et al., 2017), Advantage Actor Critic (A2C) (Mnih et al., 2016), and Deep Q-Networks (DQN) (Mnih et al., 2015), using the OpenAI Baselines implementations (Hesse et al., 2017). Both PPO and A2C find it challenging to learn the harder difficulty settings of the game, and perform better with deeper networks (Figure 2). DQN performs surprisingly well, but we see some improvement in performance variance with a deeper model. In summary, while the policy can theoretically be expressed with a linear model, empirically we see gains in performance and a reduction in variance when using deeper networks (c.f. Figures 3, 4.)

#### 4.1.2 EVALUATING DEEPER NETWORKS

Having evaluated the performance of linear models, we try a deeper model for our policy net: a fully connected neural network with two hidden layers of width 300. (Hyperparameters were chosen without extensive tuning and by trying a few different possible settings. We found that two hidden layers generally performed best and the width of the network did not have much effect on the resutls.) Identically to above, we evaluate PPO, A2C and DQN on varying start state potentials and $K$. Each algorithm is run with 3 random seeds, and in all plots we show minimum, mean, and maximum performance. Results are shown in Figures 3, 4. Note that all algorithms show variation in performance across different settings of potentials and $K$, and show noticeable drops in performance with harder difficulty settings. When varying potential in Figure 3 both PPO and A2C show larger variance than DQN, though PPO mostly matches or beats DQN in performance. When varying $K$, PPO shows less variance than DQN. A2C shows the greatest variance and worst performance out of all three methods.

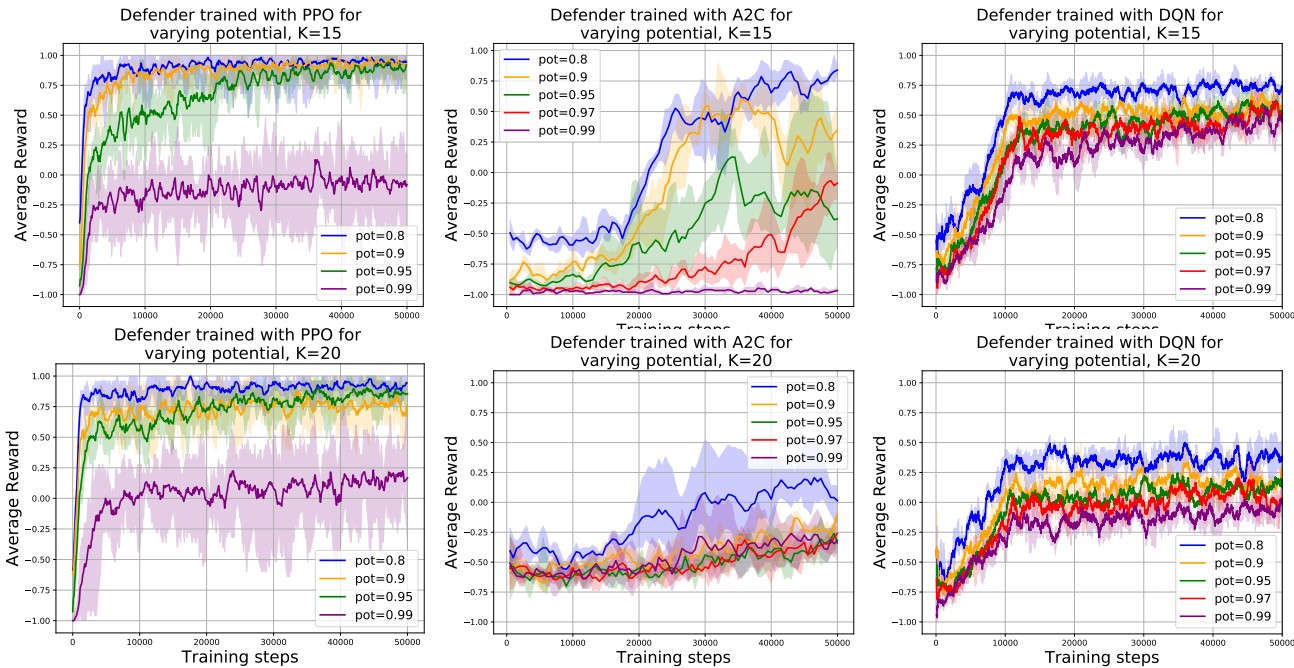

Figure 3: Training defender agent with PPO, A2C and DQN for varying values of potentials and two different choices of $K$ with a deep network. Overall, we see significant improvements over using a linear model. For smaller $K$, DQN performs relatively consistently across different potential values, though not quite matching PPO – left and right panes, row 2. A2C tends to fare worse than both PPO and DQN.

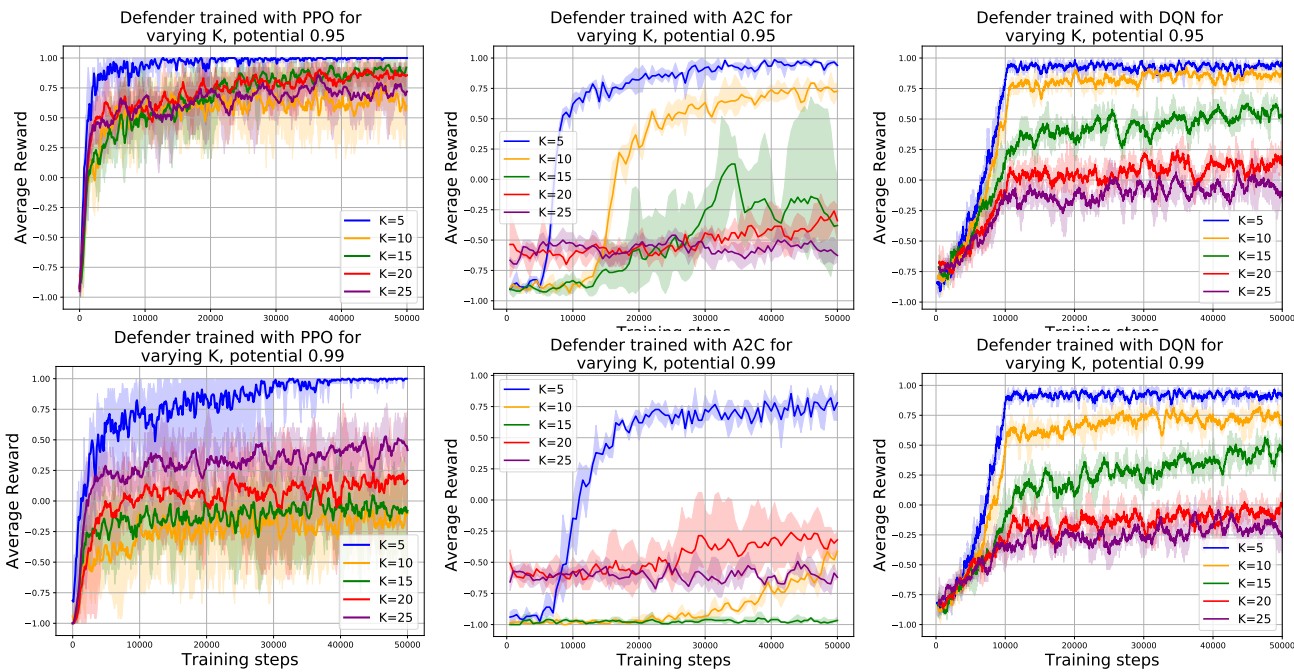

Figure 4: Training defender agent with PPO, A2C and DQN for varying values of $K$ and two different choices of potential (top and bottom row) with a deep network. All three algorithms show a noticeable variation in performance over different difficulty settings, though we note that PPO seems to be more robust to larger $K$ (which corresponds to longer episodes). A2C tends to fare worse than both PPO and DQN.

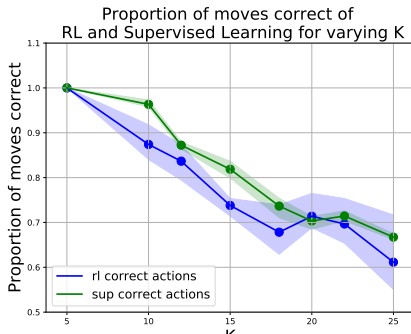 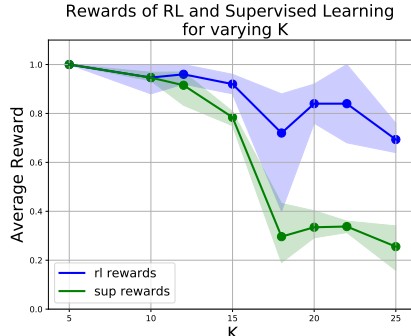

Figure 5: Plots comparing the performance of Supervised Learning and RL on the Attacker Defender Game for different choices of K. The left pane shows the proportion of moves correct for supervised learning and RL (according to the ground truth). Unsurprisingly, we see that supervised learning is better on average at getting the ground truth correct move. However, RL is better at playing the game: a policy trained through RL significantly outperforms a policy trained through supervised learning (right pane), with the difference growing for larger $K$.

## 5 SUPERVISED LEARNING VS RL

One remarkable aspect of the Attacker-Defender game is that not only do we have an easily expressible optimal policy, but we know the ground truth on a *per move* basis. We can thus compare RL to a Supervised Learning setup, where we classify the correct action on a large set of sampled states. To carry out this test in practice, we first train a defender policy with reinforcement learning, saving all observations seen to a dataset. We then train a supervised network (with the same architecture as the defender policy) to classify the optimal action. This ensures both methods see exactly the same data points. We then test the supervised network on how well it can play. The results, shown in Figure 5 are counter intuitive.

Supervised learning (unsurprisingly) has a higher proportion of correct moves: keeping count of the ground truth correct move for each turn in the game, the trained supervised policy network has a higher proportion of ground truth correct moves in play. However, despite this, reinforcement learning is better at playing the game, winning a larger proportion of games. These results are shown in Figure 5 for varying choices of $K$.

We conjecture that reinforcement learning is learning to focus most on moves that matter for winning. To investigate this conjecture, we perform two further experiments. Define a *fatal* mistake to be when the defender moves from a winning state (potential $< 1$) to a losing state (potential $> 1$) due to an incorrect move. We count the number of fatal mistakes made by the trained supervised policy, and trained RL policy. The results are shown in the left pane of Figure 6. We see that supervised learning is much more prone to make fatal mistakes, with a sharp increase in fatal mistakes for larger $K$, supporting its sharp decrease in performance.

We also look at where mistakes are made by RL and Supervised Learning based on distance of the move from the end of the game. We find that RL is better at the final couple of moves, and then consistently better in most of the earlier parts of the game.

This contrast forms an interesting counterpart to recent findings of Silver et al. (2017), who in the context of Go also compared reinforcement learning to supervised approaches. A key distinction is that their supervised work was relative to a heuristic objective, whereas in our domain we are able to compare to provably optimal play.

## 6 GENERALIZATION AND MULTIAGENT LEARNING

Returning to our RL Defender Agent, we would like to know how robust its learned policy is. In particular, as we have so far been training our agent with a randomized but hard coded attacker, we would like to test how sensitive a defender agent is to the particular attacker strategy. We investigate this in Figure 7 where we first train a defender agent on the optimal attacker and test on the disjoint support attacker. We notice a large drop in performance when switching from the optimal attacker to

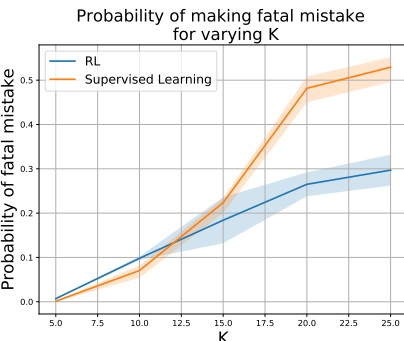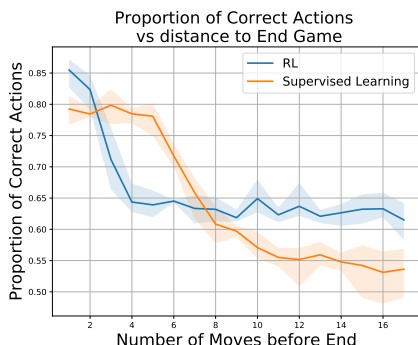

Figure 6: Figure showing: (1) (left pane) the number of fatal mistakes (defender moves from winning state (less than 1.0 potential) to losing state (greater than 1.0 potential)) made by supervised learning compared to RL. We find that Supervised Learning makes many more fatal mistakes, explaining its collapse in performance. (2) (right pane) plot showing when (measured as distance to end game) RL and supervised learning make mistakes. RL is more accurate than supervised learning at predicting the right action for the final couple of moves, and then drops quickly to a constant, whereas supervised learning is less accurate right at the very end and drops more slowly but much further, having lower accuracy than RL for many of the earlier moves.

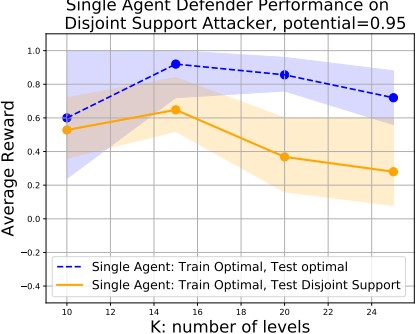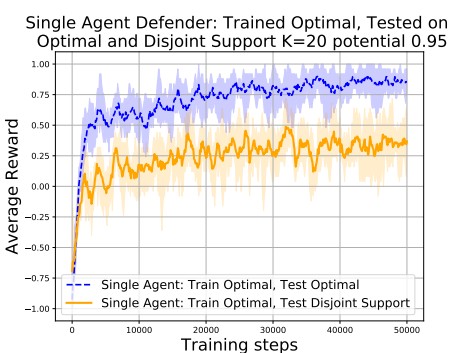

Figure 7: Plot showing overfitting to opponent strategies. A defender agent is trained on the optimal attacker, and then tested on (a) another optimal attacker environment (b) the disjoint support attacker environment. The left pane shows the resulting performance drop when switching to testing on the same opponent strategy as in training to a different opponent strategy. The right pane shows the result of testing on an optimal attacker vs a disjoint support attacker during training. We see that performance on the disjoint support attacker converges to a significantly lower level than the optimal attacker.

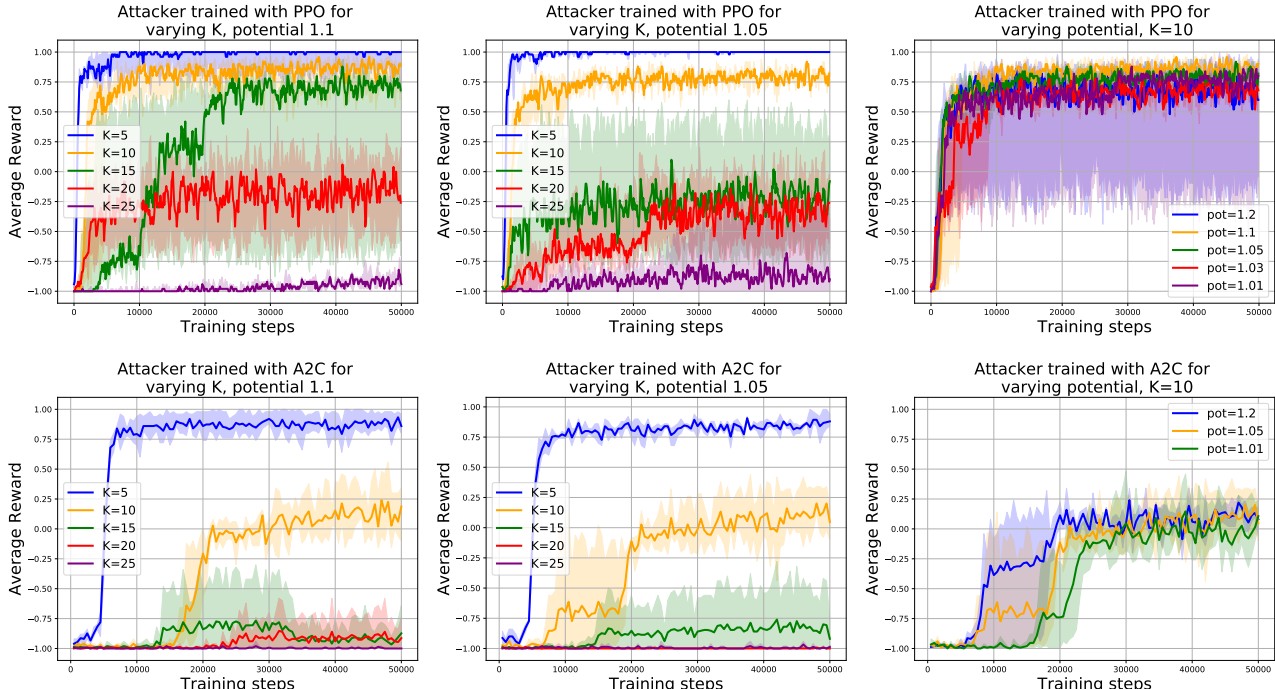

Figure 8: Performance of PPO and A2C on training the attacker agent for different difficulty settings. DQN performance was very poor (reward $< -0.8$ at $K = 5$ with best hyperparams). We see much greater variation of performance with changing $K$, which now affects the sparseness of the reward as well as the size of the action space. There is less variation with potential, but we see a very high performance variance (top right pane) with lower (harder) potentials.

the disjoint support attacker. As we know there exists an optimal policy which generalizes perfectly across all attacker strategies, this result suggests that the defender is overfitting to the particular attacker strategy.

## 6.1 TRAINING AN ATTACKER AGENT

One way to mitigate this overfitting issue is to set up a method of also training the attacker, with the goal of training the defender against a learned attacker, or even better, in the multiagent setting. However, determining the correct setup to train the attacker agent first requires devising a tractable parametrization of the action space. A naive implementation of the attacker would be to have the policy output how many pieces should be allocated to $A$ for each of the $K + 1$ levels (as described in Spencer (1994)). This can grow exponentially in $K$, which is clearly impractical. To address this, we first prove a theorem that enables us to show that we can parametrize an optimal attacker with a much smaller action space.

**Theorem 3.** *For any Attacker-Defender game with $K$ levels, start state $S_0$ and $\phi(S_0) \geq 1$, there exists a partition $A, B$ such that $\phi(A) \geq 0.5$, $\phi(B) \geq 0.5$, and for some $l$, $A$ contains pieces of level $i > l$, and $B$ contains all pieces of level $i < l$.*

*Proof.* For each $l \in \{0, 1, \ldots, K\}$, let $A_l$ be the set of all pieces from levels $K$ down to and excluding level $l$, with $A_K = \emptyset$. We have $\phi(A_{i+1}) \leq \phi(A_i)$, $\phi(A_K) = 0$ and $\phi(A_0) = \phi(S_0) \geq 1$. Thus, there exists an $l$ such that $\phi(A_l) < 0.5$ and $\phi(A_{l-1}) \geq 0.5$. If $\phi(A_{l-1}) = 0.5$, we set $A_{l-1} = A$ and $B$ the complement, and are done. So assume $\phi(A_l) < 0.5$ and $\phi(A_{l-1}) > 0.5$

Since $A_{l-1}$ only contains pieces from levels $K$ to $l$, potentials $\phi(A_l)$ and $\phi(A_{l-1})$ are both integer multiples of $2^{-(K-l)}$, the value of a piece in level $l$. Letting $\phi(A_l) = n \cdot 2^{-(K-l)}$ and $\phi(A_{l-1}) = m \cdot 2^{-(K-l)}$, we are guaranteed that level $l$ has $m - n$ pieces, and that we can move $k < m - n$ pieces from $A_{l-1}$ to $A_l$ such that the potential of the new set equals 0.5. $\square$

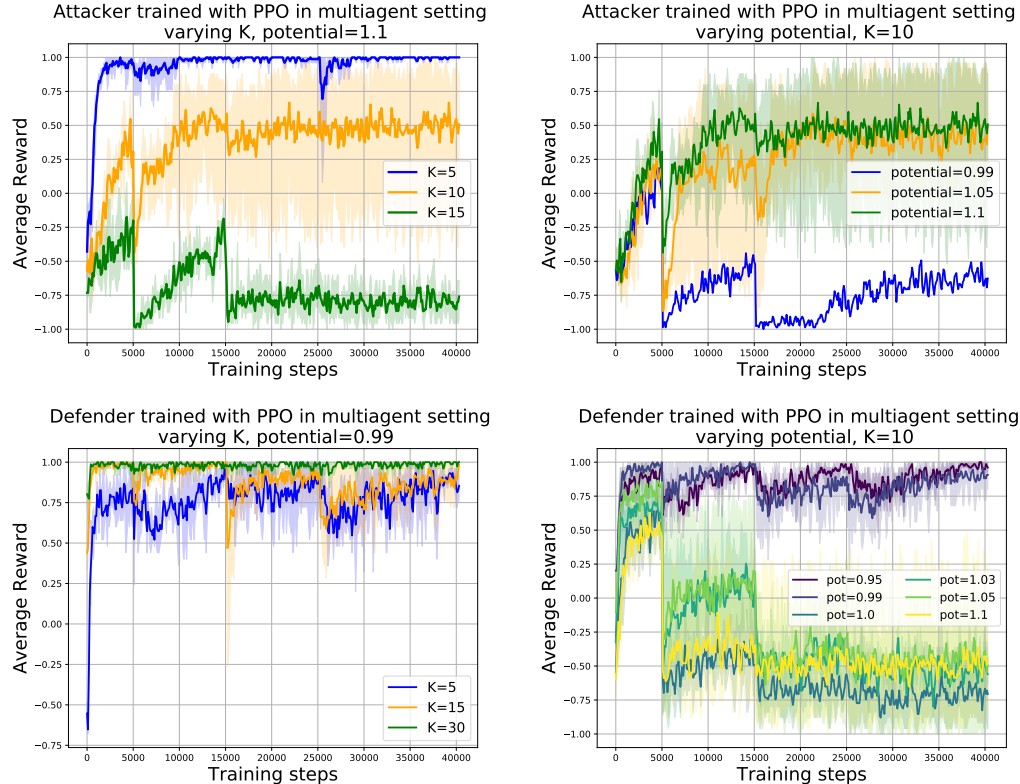

Figure 9: Performance of attacker and defender agents when learning in a multiagent setting. In the top panes, solid lines denote attacker performance. In the bottom panes, solid lines are defender performance. The sharp changes in performance correspond to the times we switch which agent is training. We note that the defender performs much better in the multiagent setting: comparing the top and bottom left panes, we see far more variance and lower performance of the attacker compared to the defender performance below. Furthermore, the attacker loses to the defender for potential $1.1$ at $K = 15$, despite winning against the optimal defender in Figure 8. We also see (right panes) that the attacker has higher variance and sharper changes in its performance even under conditions when it is guaranteed to win.

This theorem gives a different attacker parametrization. The attacker outputs a level $l$. The environment assigns all pieces before level $l$ to $A$, all pieces after level $l$ to $B$, and splits level $l$ among $A$ and $B$ to keep the potentials of $A$ and $B$ as close as possible. Theorem 3 guarantees the optimal policy is representable, and the action space linear in $K$ instead of exponential in $K$.

With this setup, we train an attacker agent against the optimal defender with PPO, A2C, and DQN. The DQN results were very poor, and so we show results for just PPO and A2C. In both algorithms we found there was a large variation in performance when changing $K$, which now affects both reward sparsity and action space size. We observe less outright performance variability with changes in potential for small $K$ but see an increase in the variance (Figure 8).

## 6.2 LEARNING THROUGH MULTIAGENT PLAY

With this attacker training, we can now look at learning in a multiagent setting. We first explore the effects of varying the potential and $K$ as shown in Figure 9. Overall, we find that the attacker fares *worse* in multiagent play than in the single agent setting. In particular, note that in the top left pane of Figure 9, we see that the attacker loses to the defender even with $\phi(S_0) = 1.1$ for $K = 15$. We can compare this to Figure 8 where with PPO, we see that with $K = 15$, and potential $1.1$, the single agent attacker succeeds in winning against the optimal defender.

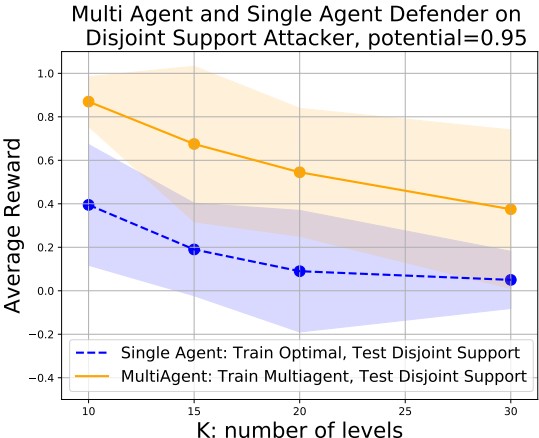

Figure 10: Results for generalizing to different attacker strategies with single agent defender and multiagent defender. The figure single agent defender trained on the optimal attacker and then tested on the disjoint support attacker and a multiagent defender also tested on the disjoint support attacker for different values of $K$. We see that multiagent defender generalizes better to this unseen strategy than the single agent defender.

### 6.3 SINGLE AGENT AND MULTIAGENT GENERALIZATION ACROSS OPPONENT STRATEGIES

Finally, we return again to our defender agent, and test generalization between the single and multiagent settings. We train a defender agent in the single agent setting against the optimal attacker, and test on a an attacker that only uses the Disjoint Support strategy. We also test a defender trained in the multiagent setting (which has never seen any hardcoded strategy of this form) on the Disjoint Support attacker. The results are shown in Figure 10. We find that the defender trained as part of a multiagent setting generalizes noticeably better than the single agent defender. We show the results over 8 random seeds and plot the mean (solid line) and shade in the standard deviation.

## 7 CONCLUSION

In this paper, we have proposed Erdos-Selfridge-Spencer games as rich environments for investigating reinforcement learning, exhibiting continuously tunable difficulty and an exact combinatorial characterization of optimal behavior. We have demonstrated that algorithms can exhibit wide variation in performance as we tune the game's difficulty, and we use the characterization of optimal behavior to expose intriguing contrasts between performance in supervised learning and reinforcement learning approaches. Having reformulated the results to enable a trainable attacker, we have also been able to explore insights on overfitting, generalization, and multiagent learning. We also develop further results in the Appendix, including an analysis of catastrophic forgetting, generalization across different values of the game's parameters, and a method for investigating measures of the model's confidence. We believe that this family of combinatorial games can be used as a rich environment for gaining further insights into deep reinforcement learning.

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

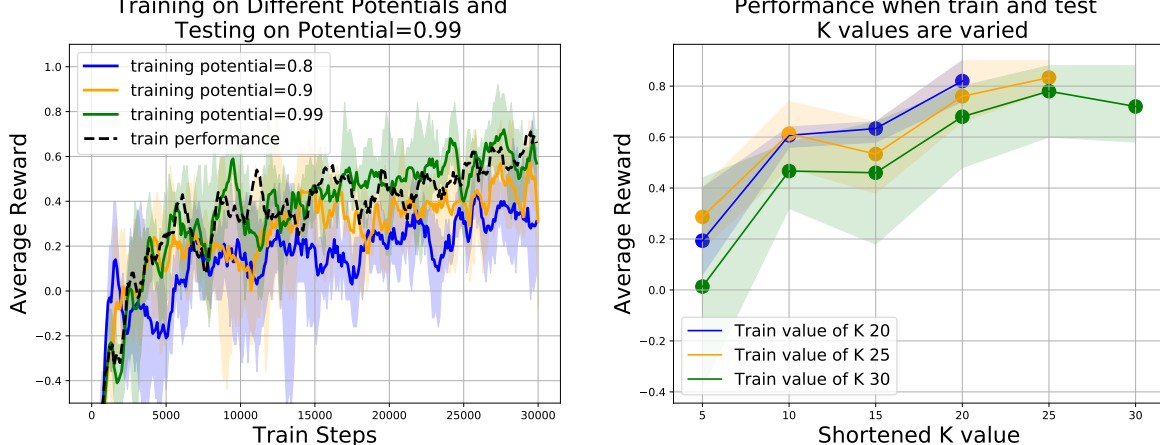

Figure 11: On the left we train on different potentials and test on potential $0.99$. We find that training on harder games leads to better performance, with the agent trained on the easiest potential generalizing worst and the agent trained on a harder potential generalizing best. This result is consistent across different choices of test potentials. The right pane shows the effect of training on a larger $K$ and testing on smaller $K$. We see that performance appears to be inversely proportional to the difference between the train $K$ and test $K$.

# Appendix

## Other Generalization Phenomena

### Generalizing Over Different Potentials and $K$

In the main text we examined how our RL defender agent performance varies as we change the difficulty settings of the game, either the potential or $K$. Returning again to the fact that the Attacker-Defender game has an expressible optimal that generalizes across all difficulty settings, we might wonder how training on one difficulty setting and testing on a different setting perform. Testing on different potentials in this way is straightforwards, but testing on different $K$ requires a slight reformulation. our input size to the defender neural network policy is $2(K + 1)$, and so naively changing to a different number of levels will not work. Furthermore, training on a smaller $K$ and testing on larger $K$ is not a fair test – the model cannot be expected to learn how to weight the lower levels. However, testing the converse (training on larger $K$ and testing on smaller $K$) is both easily implementable and offers a legitimate test of generalization. We find (a subset of plots shown in Figure 11) that when varying potential, training on harder games results in better generalization. When testing on a smaller $K$ than the one used in training, performance is inverse to the difference between train $K$ and test $K$.

### Catastrophic Forgetting and Curriculum Learning

Recently, several papers have identified the issue of catastrophic forgetting in Deep Reinforcement Learning, where switching between different tasks results in destructive interference and lower performance instead of positive transfer. We witness effects of this form in the Attacker-Defender games. As in Section 7, our two environments differ in the $K$ that we use – we first try training on a small $K$, and then train on larger $K$. For lower difficulty (potential) settings, we see that this curriculum learning improves play, but for higher potential settings, the learning interferes catastrophically, Figure 12

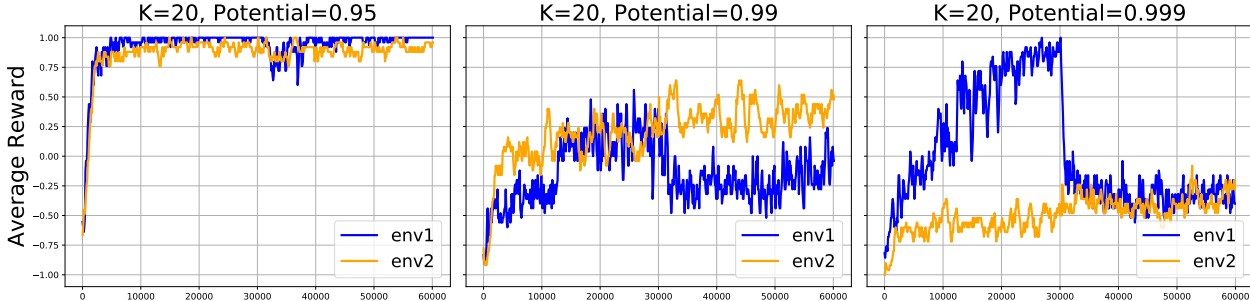

Figure 12: Defender agent demonstrating catastrophic forgetting when trained on environment 1 with smaller $K$ and environment 2 with larger $K$.

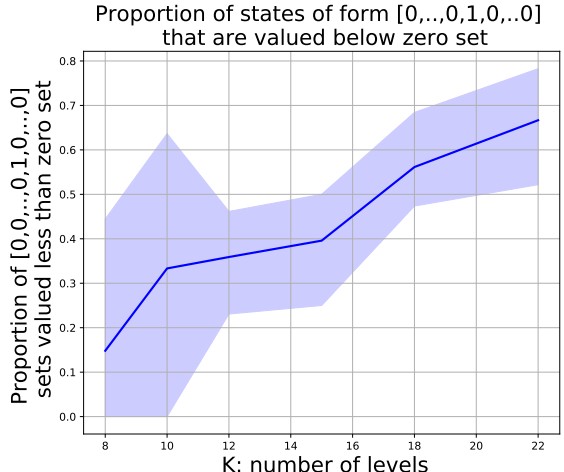

Figure 13: Figure showing proportion of sets of form $[0, ..., 0, 1, 0, ..., 0]$ that are valued less than the null set. Out of the $K + 1$ possible one hot sets, we determine the proportion that are not picked when paired with the null (zero) set, and plot this value for different $K$.

## UNDERSTANDING MODEL FAILURES

### VALUE OF THE NULL SET

The significant performance drop we see in Figure 7 motivates investigating whether there are simple rules of thumb that the model has successfully learned. Perhaps the simplest rule of thumb is learning the value of the null set: if one of $A, B$ (say $A$) consists of only zeros and the other ($B$) has some pieces, the defender agent should reliably choose to destroy $B$. Surprisingly, even this simple rule of thumb is violated, and even more frequently for larger $K$, Figure 13.

### MODEL CONFIDENCE

We can also test to see if the model outputs are well *calibrated* to the potential values: is the model more confident in cases where there is a large discrepancy between potential values, and fifty-fifty where the potential is evenly split? The results are shown in Figure 14.

### 7.1 GENERALIZING ACROSS START STATES AND OPPONENT STRATEGIES

In the main paper, we mixed between different start state distributions to ensure a wide variety of states seen. This begets the natural question of how well we can generalize across start state distribution if we train on purely one distribution. The results in Figure 15 show that training naively

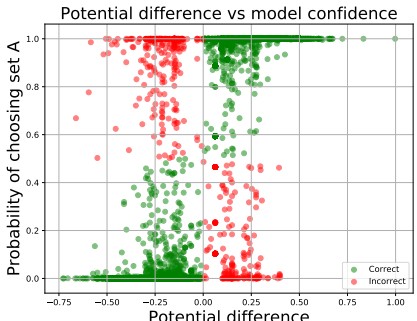 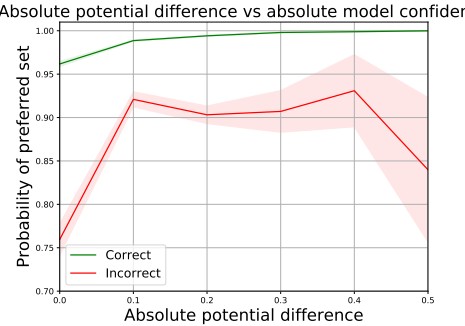

Figure 14: Confidence as a function of potential difference between states. The top figure shows true potential differences and model confidences; green dots are moves where the model prefers to make the right prediction, while red moves are moves where it prefers to make the wrong prediction. The right shows the same data, plotting the *absolute* potential difference and absolute model confidence in its preferred move. Remarkably, an increase in the potential difference associated with an increase in model confidence over a wide range, even when the model is wrong.

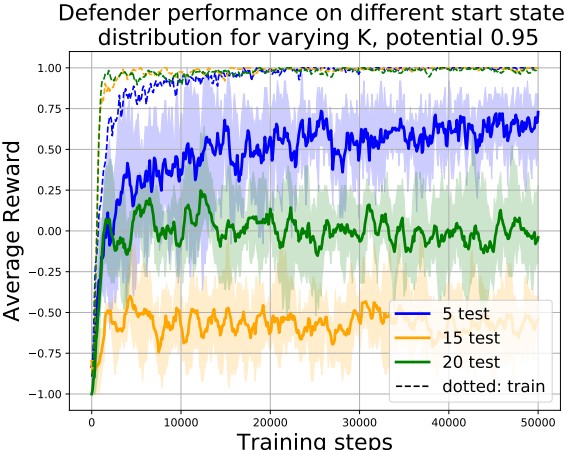 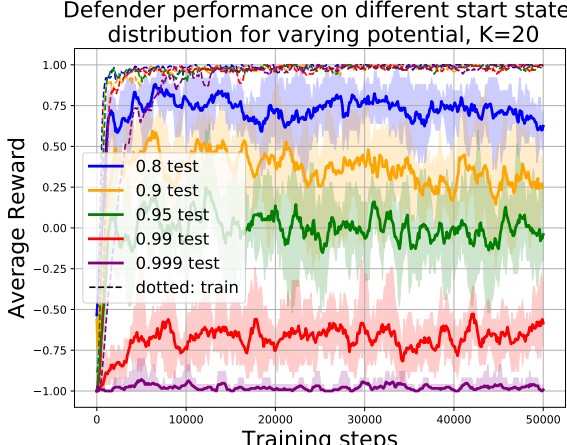

Figure 15: Change in performance when testing on different state distributions

on an 'easy' start state distribution (one where most of the states seen are very similar to one another) results in a significant performance drop when switching distribution.

In fact, the amount of possible starting states for a given $K$ and potential $\phi(S_0) = 1$ grows super exponentially in the number of levels $K$. We can state the following theorem:

**Theorem 4.** *The number of states with potential* 1 *for a game with $K$ levels grows like* $2^{\Theta(K^2)}$ *(where* $0.25K^2 \leq \Theta(K^2) \leq 0.5K^2$ *)*

We give a sketch proof.

*Proof.* Let such a state be denoted $S$. Then a trivial upper bound can be computed by noting that each $s_i$ can take a value up to $2^{(K-i)}$, and producting all of these together gives roughly $2^{K/2}$.

For the lower bound, we assume for convenience that $K$ is a power of 2 (this assumption can be avoided). Then look at the set of non-negative integer solutions of the system of simultaneous equations

$$a_{j-1}2^{1-j} + a_j2^{-j} = 1/K$$

where j ranges over all even numbers between $\log(K) + 1$ and $K$. The equations don't share any variables, so the solution set is just a product set, and the number of solutions is just the product

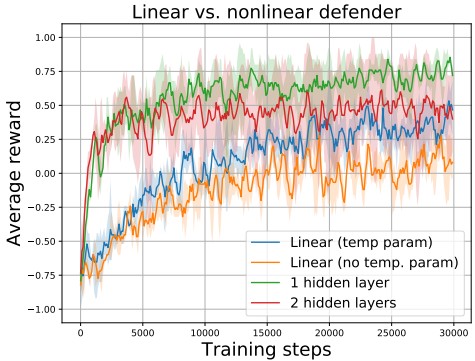

Figure 16: Effect of model size on performance for $K = 10$. Hidden layers have 300 units.

$\prod_j (2^{j-1}/K)$ where, again, $j$ ranges over even numbers between $\log(K) + 1$ and $K$. This product is roughly $2^{K^2/4}$. $\qquad\square$

## VARYING MODEL DEPTH

As the optimal defender policy is expressible as a linear model, we empirically investigate whether depth is helpful. We find that even with a temperature included, linear models perform worse than models with one or two hidden layers.

