# OpenReview forum: "Can Deep Reinforcement Learning solve Erdos-Selfridge-Spencer Games?"
_ICLR.cc/2018/Conference — Invite to Workshop Track_

### Official Review · AnonReviewer3 · 2017-11-23
**The paper is interesting and germane, but there are some clarity issues with the justification and evaluation which undermine the message the authors are trying to make.**

**Rating:** 5
**Confidence:** 3

**Review:**

The paper presents Erdos-Selfridge-Spencer games as environments for investigating
deep reinforcement learning algorithms. The proposed games are interesting and clearly challenging, but I am not sure what they tell us about the algorithms chosen to test them. There are some clarity issues with the justification and evaluation which undermine the message the authors are trying to make.

In particular, I have the following concerns:

  • these games have optimal policies that are expressible as a linear model, meaning that if the architecture or updating of the learning algorithm is such that there is a bias towards exploring these parts of policy space, then they will perform better than more general algorithms. What does this tell us about the relative merits of each approach? The authors could do more to formally motivate these games as "difficult" for any deep learning architecture if possible.
  • the authors compare linear models with non-linear models at some point for attacker policies, but it is unclear whether these linear models are able to express the optimal policy. In fact, there is a level of non-determinism in how the attacker policies are encoded which means that an optimal policy cannot be (even up to soft-max) expressed by the agent (as I read things the number of pieces chosen in level l is always chosen uniformly randomly).
  • As the authors state, this paper is an empirical evaluation, and the theorems presented are derived from earlier work. There is possibly too much focus on the proofs of these theorems.
  • There are a number of ambiguities and errors which places difficulties on the interpretation (and potential replication) of the experiments. As this is an empirical study, this is the yardstick by which the paper should be judged. In particular, this relates to:
    ◦ The architecture of each of the tested Deep RL methods.
    ◦ What is done to select appropriate tuning parameters of the tested Deep RL methods, if anything.
    ◦ It is unclear whether 'incorrect actions' in the supervised learning evaluations, refer to non-optimal actions, or simply actions that do not preserve the dominance of the defender, e.g. both partitions may have potential >0.5
    ◦ Fig 4. right looks like a reward signal, but is labelled Proportion correct. The text is not clear enough to be sure which it is.
    ◦ Fig 4. left and right has 4 methods: rl rewards, rl correct actions, sup rewards, and sup correct actions. The specifics of how these methods are constructed is unclear from the paper.
    ◦ What parts of the evaluation explores how well these methods are able to represent the states (feature/representation learning) and what parts are evaluating the propagation of sparse rewards (the reinforcment learning core)? The authors could be clearer and more targetted with respect to this question.

There is value in this work, but in its current state I do not think it is ready for publicaiton.

# Detailed notes

[p4, end of sec 3] The authors say that the difficulty of the games can be varied with "continuous changes in potential", but the potential is derived from the discrete initial game state, so these values are not continuously varying (even though it is possible to adjust them by non-integer amounts).

[p4, sec 4.1]
"strategy unevenly partitions the occupied levels...with the proportional difference between the two sets being sampled randomly"
What is meant by this? The proportional difference between the two sets is discussed as if it is a continuous property, but must be chosen from the discrete set of all available partitions. If one partition one is chosen uniformly randomly from all possibly sets A, B (and the potential proportion calculated) then I don't know why it would be written in this way. That suggests that proportions that are closer to 1:1 are chosen more often than "extreme" partitions, but how? This feels a little under-justified.
"very different states A, B (uneven potential, disjoint occupied levels)"
Are these states really "very different", or at least for the reasons indicated. Later on (Theorem 3) we see how an optimal partition is generated. This chooses a partition where one part contains all pieces in layer (l+1) and above and one part with all pieces in layer (l-1) and below, with layer l being distributed between the two parts. The first part will typically have a slightly lower potential than the other and all layers other than layer l will be disjoint.


[p6, Fig 4] The right plot y-limits vary between -1 and 1 so it cannot represent a proportion of correct actions. Also, in the text the authors say:
  >> The results, shown in Figure 4 are surprising. Reinforcement learning
  >> is better at playing the game, but does worse at predicting optimal moves.
I am not sure which plot shows the playing of the game. Is this the right hand plot? In which case are we looking at rewards? In fact, I am a little confused as to what is being shown here. Is "sup rewards" a supervised learning method trained on rewards, or evaluated on rewards, or both? And how is this done. The text is just not clear enough.

[p7 Fig 6 and text] Here the authors are comparing how well agents select the optimal actions as compared to how close they are to the end of the game. This relates to the "surprising" fact that "Reinforcement learning is better at playing the game, but does worse at predicting optimal moves.". I think an important point here is how many training/test examples there are in each bin. If there are more in the range 3-7 moves from the end of the game, than there are outside this range, then the supervised learner will

[p8 proof of theorem 3]
"φ(A l+1 ) < 0.5 and φ(A l ) > 0.5."
Is it true that both these inequalities are strict?
"Since A l only contains pieces from levels K to l + 1"
In fact this should read from levels K to l.
"we can move k < m − n pieces from A l+1 to A l"
Do the authors mean that we can define a partition A, B where A = A_{l+1} plus some (but not all) elements in level l (A_{l}\setminus A_{l+1})?
"...such that the potential of the new set equals 0.5"
It will equal exactly 0.5 as suggested, but the authors could make it more precise as to why (there is a value n+k < l (maybe <=l) such that (n+k)*2^{-(K-l+1)}=0.5 (guaranteed). They should also indicate why this then justifies their proof (namely that phi(S0)-0.5 >= 0.5).

[p8 paramterising action space] A comment: this doesn't give as much control as the authors suggest. Perhaps the agent should also chose the proportion of elements in layer l to set A. For instance, if there are a large number of elements in l, and or phi(A_{l+1}) is very close to 0.5 (or phi(A_l) is very close to 0.5) then this doesn't give the attacker the opportunity to fine tune the policy to select very good partitions.  It is unclear expected level of control that agents have under various conditions (K and starting states).

[p9 Fig 8] As the defender's score is functionally determined by the attackers score, it doesn't help to include this on the plot. It just distracts from the signal.

---

> ### Author Response · Authors · 2017-12-17
> **Response to Review 3: Part 1**
>
> Thank you for your time in reviewing the paper and your comments! We’ve uploaded a new version of the paper based on the feedback, and have addressed specific points below.
>
> ##### “Optimal policies expressible as a linear model” #####
> We have added a linear baseline (Figure 2 and Section 4.1.1) showing that these games are hard to learn well with what is theoretically an expressive enough model. This is additional motivation for studying deeper architectures.
>
> #### “'incorrect actions' in the supervised learning evaluations” ####
> The natural formulation of the optimal policy is for the agent to keep the potential in the next state reached as small as possible; this ensures that the agent preserves the minimax value of the game from all states. This policy corresponds to choosing the set of larger potential to delete; correspondingly, an incorrect action is one where the agent does not choose the set of larger potential. In the supervised learning setting, we have a starting potential < 1, so the defender, if playing according to the optimal policy, is guaranteed a win and one of A, B will always have potential < 0.5. Even if, due to suboptimal play, there was a state where A, B both have potential > 0.5, the policy that minimizes the potential in the next state would still have a well-defined move, which is to choose the set of larger potential to delete; we would view this as the correct move under optimal play.
>
> ##### Figure 4 (in old version) now Figure 5 ####
> Setup for Figure 4: we first train a RL defender agent to play the game, and store all the game trajectories that it sees. We then take each state in the game, and label it with the correct action (as described in the comment above, i.e. the correct action is picking the `larger’ set to destroy, which is what the optimal policy does.) We train a model in a supervised fashion on this labelled dataset
>
> We now have edited Figure 4 based on your feedback to make it clearer (it is Figure 5 in the new version.) The left pane shows the proportion of correct actions for different K achieved by RL and Supervised Learning.
>
> Unsurprisingly, we see that supervised learning is better than RL in terms of number of correct actions. However, RL is better at playing the game: we take a model trained in (1) supervised fashion (2) with RL and test it on the environment, and find that RL achieves significantly higher reward, particularly for larger K.
>
> We investigate this further in Figure 6 with a new plot (left pane), through studying “fatal mistakes” -- errors made that take the agent from a winning state to a losing state. We find that Supervised Learning is much more prone to fatal mistakes than RL, suggesting a natural basis for the worse performance.
>
> ##### Hyperparameter Choice for Deep RL architectures #####
> Aside from experiments to determine the effect of depth and width of architectures, we used minimal hyperparameter tuning. While additional hyperparameter tuning would have likely helped improve performance, the overall conclusions drawn from the paper (better generalization in multiagent vs single agent, ability of RL to avoid “fatal mistakes” made by supervised learning, decrease in rewards as game difficulty increased) would not have been affected by hyperparameter changes.
>
> ##### “ As the authors state, this paper is an empirical evaluation, and the theorems presented are derived from earlier work” #####
> This is not completely the case: Theorem 3 is a theoretical contribution that is original to this paper; and it is an important component of the paper since without it, training an attacker agent would be intractable. The earlier theorems are explained in detail since the central approach of the paper is based on the linearly expressible potential function and its connection to the optimal policy, and one needs the proofs of these earlier theorems -- not simply their statements -- in order to understand this structure.
>
> ##### “The authors compare linear models with non-linear models for attacker policies” #####
> This is incorrect -- we don’t use linear models for attacker policies.
>
> #### [p4, end of sec 3] Continuous changes of potential ####
> The potential changes are indeed due to the discrete initial game state, but for a game with K levels, we can adjust the potential in increments of 2^-K (e.g. for K=20, we can adjust the potential in increments of ~0.0000009) which seems to be a reasonable approximation to continuous.

---

> > ### Author Response · Authors · 2017-12-17
> > **Response to Review 3, Part 2**
> >
> > ####” [p4, sec 4.1]
> > "strategy unevenly partitions the occupied levels...with the proportional difference between the two sets being sampled randomly"” ######
> > To clarify the method: we randomly pick a proportion C that is **bounded away*** from 0.5, and let the potential of the first set be C*potential. We then greedily fill up a set until its potential first crosses C*potential. The remainder of the pieces go to the other set. The states are typically of significantly different potentials when sampled this way, due to the ability to increment by very small amounts (amounts of 2^-K).
> >
> > ##### Later on (Theorem 3) we see how an optimal partition is generated….The first part will typically have a slightly lower potential than the other and all layers other than layer l will be disjoint. #####
> > In Theorem 3, we show that there is a way to form the partition A, B with **almost** disjoint support: as you’ve written above, A will contain all pieces from (l+1) up, and B from (l-1) and below, ***but*** they might share pieces in level l (with the optimal splitting done by the environment.) As a result, there is no bias towards the first set having a slightly lower potential.
> >
> > ##### [p7 Fig 6 and text] ######
> > It looks like your comment here is unfinished? More than data being in different bins, the important aspect to make the comparison fair is that the RL agent and the Supervised Model see exactly the same data, which we ensure by first generating the data with the RL agent interacting with the environment, and using that data for supervised training.
> >
> > ##### [p8 proof of theorem 3]  #####
> > Thank you for the questions about the proof, we’ve corrected the indexing typo and we hope the argument is clearer now (in the v2 uploaded!) We’d be happy to take additional questions on this.
> >
> > ##### [p9 Fig 8] #####
> > We’ve edited the figure to remove the dashed lines which hopefully makes the curves clearer.

---

> ### Author Response · Authors · 2018-01-03
> **Response to Revision and Rebuttal?**
>
> Dear Reviewer,
>
> Happy new year! We would be very grateful to know your thoughts on our paper revision and rebuttal, which we hope has answered the points raised.
>
> Best,
>
> The Authors

---

> ### Comment · AnonReviewer3 · 2018-01-04
> **In response to the rebuttal**
>
> Thank you for your thoughtful responses. I have tried to respond to each below. I think that the work is interesting, but I am keeping my recommendation as weak reject. In particular, the first two comments below indicate the issues I consider most problematic.
>
> >> ##### “Optimal policies expressible as a linear model” #####
> >> We have added a linear baseline (Figure 2 and Section 4.1.1) showing that these games are hard to learn well with what is theoretically an expressive enough model. This is additional motivation for studying deeper architectures.
> >>
>
> Just to be precise, it doesn't show that the games are hard to learn with any method. It shows that the current methods do not learn well on these games. The fact that there is an optimal policy based on a linear function of the input space means that a deep learner isn't actually needed. The authors could do more in the paper to give some intuition to why these policies are hard to learn. Is it because the weights of the optimal policy are so different in terms of magnitude? If so, perhaps a linear model with the weights trained in the log domain would suffice to learn these games efficiently. Also, what features are the deep learners actually learning that enables them to improve beyond that of a linear learner? Some investigation of that would help.
>
> I remain unconvinced that these games are good general tests for deep reinforcment learning. I think this would require more theoretical justification of why a deep learner (or shallow learner) simply cannot learn them efficiently, and I am not sure that is possible.
>
>
> >> #### “'incorrect actions' in the supervised learning evaluations” ####
> >> The natural formulation of the optimal policy is for the agent to keep the potential in the next state reached as small as possible; this ensures that the agent preserves the minimax value of the game from all states. This policy corresponds to choosing the set of larger potential to delete; correspondingly, an incorrect action is one where the agent does not choose the set of larger potential. In the supervised learning setting, we have a starting potential < 1, so the defender, if playing according to the optimal policy, is guaranteed a win and one of A, B will always have potential < 0.5. Even if, due to suboptimal play, there was a state where A, B both have potential > 0.5, the policy that minimizes the potential in the next state would still have a well-defined move, which is to choose the set of larger potential to delete; we would view this as the correct move under optimal play.
> >>
>
> Under these circumstances both moves can lead to success, and so both are optimal. To put this another way, a perfect player (one that never lost when it could win) could chose the set with the lower potential under these conditions and still win every time. The loss function in the RL domain is whether the game is won or lost, while the loss function in your supervised learning evaluation is different. While this doesn't mean the two can be compared, I would recommend discussing the meaning of this with more care.
>
>
> >> ##### Figure 4 (in old version) now Figure 5 ####
> >> Setup for Figure 4: we first train a RL defender agent to play the game, and store all the game trajectories that it sees. We then take each state in the game, and label it with the correct action (as described in the comment above, i.e. the correct action is picking the `larger’ set to destroy, which is what the optimal policy does.) We train a model in a supervised fashion on this labelled dataset
> >>
> >> We now have edited Figure 4 based on your feedback to make it clearer (it is Figure 5 in the new version.) The left pane shows the proportion of correct actions for different K achieved by RL and Supervised Learning.
> >>
> >> Unsurprisingly, we see that supervised learning is better than RL in terms of number of correct actions. However, RL is better at playing the game: we take a model trained in (1) supervised fashion (2) with RL and test it on the environment, and find that RL achieves significantly higher reward, particularly for larger K.
> >>
>
> Again, this relates to the loss function. In the supervised learning case, you are penalising "incorrect" actions uniformly, whereas in the reinforcement learning case the learner will place more emphasis on some actions than others.
>
> There is a related notion that the best action in the context of an optimal player, may not always be the best action in the context of a suboptimal player. Your notion of incorrect action assumes that you are playing an optimal player. It may be worth making the adversarial nature of the domain more explicit.

---

> > ### Comment · AnonReviewer3 · 2018-01-04
> > **In response to the rebuttal (cont)**
> >
> > >> We investigate this further in Figure 6 with a new plot (left pane), through studying “fatal mistakes” -- errors made that take the agent from a winning state to a losing state. We find that Supervised Learning is much more prone to fatal mistakes than RL, suggesting a natural basis for the worse performance.
> > >>
> >
> > Yes, this is a helpful piece of analysis.
> >
> > >> ##### Hyperparameter Choice for Deep RL architectures #####
> > >> Aside from experiments to determine the effect of depth and width of architectures, we used minimal hyperparameter tuning. While additional hyperparameter tuning would have likely helped improve performance, the overall conclusions drawn from the paper (better generalization in multiagent vs single agent, ability of RL to avoid “fatal mistakes” made by supervised learning, decrease in rewards as game difficulty increased) would not have been affected by hyperparameter changes.
> > >>
> >
> > This is a very strong claim. I am not sure I am able to to assess its validity without a theoretical justification or empirical evidence that hyperparameter tuning has no qualitative effect.
> >
> >
> > >> ##### “ As the authors state, this paper is an empirical evaluation, and the theorems presented are derived from earlier work” #####
> > >> This is not completely the case: Theorem 3 is a theoretical contribution that is original to this paper; and it is an important component of the paper since without it, training an attacker agent would be intractable. The earlier theorems are explained in detail since the central approach of the paper is based on the linearly expressible potential function and its connection to the optimal policy, and one needs the proofs of these earlier theorems -- not simply their statements -- in order to understand this structure.
> > >>
> >
> > It would have helped to be clearer about this in the paper.
> >
> >
> > >> ##### “The authors compare linear models with non-linear models for attacker policies” #####
> > >> This is incorrect -- we don’t use linear models for attacker policies.
> > >>
> >
> > Thank you for the clarification.
> >
> >
> > >> #### [p4, end of sec 3] Continuous changes of potential ####
> > >> The potential changes are indeed due to the discrete initial game state, but for a game with K levels, we can adjust the potential in increments of 2^-K (e.g. for K=20, we can adjust the potential in increments of ~0.0000009) which seems to be a reasonable approximation to continuous.
> > >>
> >
> > However, it is not continuous. Inclusion of the word 'effectively' would have avoided this criticism. For the reader to understand the methods and claims, surely it is reasonable to expect precision.

---

> > > ### Comment · AnonReviewer3 · 2018-01-04
> > > **In response to the rebuttal (cont 2)**
> > >
> > > >> ####” [p4, sec 4.1] "strategy unevenly partitions the occupied levels...with the proportional difference between the two sets being sampled randomly"” ######
> > > >> To clarify the method: we randomly pick a proportion C that is **bounded away*** from 0.5, and let the potential of the first set be C*potential. We then greedily fill up a set until its potential first crosses C*potential. The remainder of the pieces go to the other set. The states are typically of significantly different potentials when sampled this way, due to the ability to increment by very small amounts (amounts of 2^-K).
> > > >>
> > >
> > > Thank you for the clarification. Does this **bounded away** notion mean that you can actually end up choosing a partition where one set has potential <0.5, even if there exists a partition where both sets have potentials >0.5? The random element suggests this is so.
> > >
> > > >> ##### Later on (Theorem 3) we see how an optimal partition is generated….The first part will typically have a slightly lower potential than the other and all layers other than layer l will be disjoint. #####
> > > >> In Theorem 3, we show that there is a way to form the partition A, B with **almost** disjoint support: as you’ve written above, A will contain all pieces from (l+1) up, and B from (l-1) and below, ***but*** they might share pieces in level l (with the optimal splitting done by the environment.) As a result, there is no bias towards the first set having a slightly lower potential.
> > > >>
> > >
> > > But these partitions are a very restricted subset of the possible (optimal) playing choices.
> > >
> > >
> > > >> ##### [p7 Fig 6 and text] ######
> > > >> It looks like your comment here is unfinished? More than data being in different bins, the important aspect to make the comparison fair is that the RL agent and the Supervised Model see exactly the same data, which we ensure by first generating the data with the RL agent interacting with the environment, and using that data for supervised training.
> > > >>
> > >
> > > You are right. Please accept my apologies. I meant to say that if there are lots of training examples containing states that appear most often 3-7 moves from the end of the game, then this will influence the performance in this region of the graph. I am now not sure that this is correct though. I think it is more likely to relate to the difference in loss function of the two approaches (as discussed above).
> > >
> > > >> ##### [p8 proof of theorem 3]  #####
> > > >> Thank you for the questions about the proof, we’ve corrected the indexing typo and we hope the argument is clearer now (in the v2 uploaded!) We’d be happy to take additional questions on this.
> > > >>
> > >
> > > Yes, this is now clearer.
> > >
> > > >> ##### [p9 Fig 8] #####
> > > >> We’ve edited the figure to remove the dashed lines which hopefully makes the curves clearer.
> > >
> > > Thank you, these are now clearer.

---

> ### Author Response · Authors · 2018-01-05
> **Jan 5th: Author Response to Reviewer's Rebuttal Response**
>
> Thank you for your detailed responses to the rebuttal. We have responses to your two main concerns, which we hope can help address the issues you raise.
>
> #### I remain unconvinced that these games are good general tests for deep reinforcement learning. I think this would require more theoretical justification of why a deep learner (or shallow learner) simply cannot learn them efficiently, and I am not sure that is possible. #####
>
> Note that the data distribution, while linearly separable, has an exponentially small margin: there can be as little as 2^{-K} difference between the two sets A, B.  In many contexts, this exponentially small margin typically results in exponentially large sample complexity for learning linear separators, e.g. the paper [1], or the lecture notes [2].
>
> [1]: Sivan Sabato, Nathan Srebro, Naftali Tishby.  Tight Sample Complexity of Large Margin Learning. Journal of Machine Learning Research 14 (2013) 2119-2149.
> [2]: https://www.cs.princeton.edu/courses/archive/fall16/cos402/lectures/402-lec4.pdf
>
> While we do not have a proof in our case, we believe that it may be possible to use similar arguments based on the exponentially small margin to try showing a lower bound that shallow learners cannot learn optimal play efficiently here as well.
>
>
> #### Under these circumstances both moves can lead to success, and so both are optimal. To put this another way, a perfect player (one that never lost when it could win) could chose the set with the lower potential under these conditions and still win every time. ####
>
> Thanks for raising this issue; we agree with your point, namely that there’s only truly a "right" answer when one of A or B has potential < 0.5 and the other has potential > 0.5. (In any other case, as you note, while it still seems the most natural to delete the set of higher potential, it doesn't actually "matter" from the point of view of preserving the minimax value in the game tree.)
>
> Based on the initial set of reviews, we took this issue into account, and have added two new results to the topic in Section 5 (on Supervised Learning vs. RL) to address the point.  Both of these new results address the question of which moves in the game we should be using to test the supervised learner relative to the RL agent.
>
> To describe these, suppose that p_A is the potential of set A, and p_B is the potential of set B, and let us rename the sets if necessary so that p_A \leq p_B.  In both results, we look only at a subsequence of moves, rather than all moves.
>
> In the first new result, left pane in Figure 6 in the revised version, we look only at moves where the current potential is < 1, but there is a “wrong move” (a “fatal mistake”) that can make the potential in the next configuration > 1.  In particular, this means that p_A + p_B < 1 (so the current position is a forced win for the defender) but p_B \geq 0.5 (so that if A is removed, this is a fatal mistake that converts the game to a forced loss for the defender).
>
> In the second new result, which we also worked out but haven’t reported in the current revision, we consider the case raised in your discussion -- the subset of moves where it matters for the minimax value which of A or B is chosen.  This corresponds to p_A < 0.5 and p_B \geq 0.5.
>
> In both cases, then, we look only at the subsequence of moves specified by the indicated predicate on potential functions, and we see which of the supervised learning algorithm or the RL agent has better performance.  The results are very similar in the two cases -- the RL agent has better performance, and significantly so as K increases.
>
> As we note in the revised Section 5, this adds to the discussion of the contrast between the supervised learning method and the RL agent.  Essentially, it suggests that while the mathematical theory of ESS games suggests a simple closed-form optimal solution -- to always delete the set of lower potential -- this is not (as you observe) the only optimal solution, since it doesn’t matter which set is deleted when max(p_A,p_B) < 0.5 or when min(p_A,p_B) \geq 0.5.  And while supervised learning is better at matching the simple closed-form solution from the discrete math literature, it performs worse than the RL agent when restricted to the subsequence of moves that actually matter.  In this way, the RL agent is performing better overall in winning the game -- while it deviates more from the strategy suggested in the mathematical literature, the strategy is actually arrives at is effective for obtaining high reward.
>
> Our revised text tries to reflect this distinction, and in a further revision we will include the second of the two additional tests above, and emphasize this point additionally in the text.

---

### Official Review · AnonReviewer1 · 2017-11-25
**interesting empirical study**

**Rating:** 6
**Confidence:** 3

**Review:**

This paper presents a study of reinforcement learning methods applied to Erdos-Selfridge-Spencer games, a particular type of two-agent, zero-sum game.  The authors describe the game and some of its properties, notably that there exists a tractable potential function that indicates optimal play for each player for every state of the board.  This is used as a sort of ground truth that enables study of the behavior of certain reinforcement learning algorithms (for just one or to both players).  An empirical study is performed, measuring the performance of both agents, tuning the difficulty of the game for each agent by changing the starting position of the game.

- The comparison of supervised learning vs RL performance is interesting.  Is the supervised algorithm only able to implement Markovian policies?  Is the RL agent able to find policies with longer-term dependence that it can follow?  Is that what is meant by the sentence on page 6 "We conjecture that reinforcement learning is learning to focus most on moves that matter for winning"?

- Why do you think the defender trained as part of a multiagent setting generalizes better than the single agent defender?  Is there something different about the distribution of policies seen by each defender?

Quality: The method appears to be technically correct, clearly written, and easy to read.

Originality: I believe this is the first use of ESS games to study RL algorithms.  I am also not aware of trying to use games with known potential functions/optimal moves as a way to study the performance of RL algorithms.

Impact: I think this is an interesting and creative contribution to studying RL, particularly the use of an easy-to-analyze game in an RL setting.

---

> ### Author Response · Authors · 2017-12-17
> **Response to Review 2**
>
> Thank you for your time in reviewing the paper and your comments! We’ve uploaded a new version of the paper based on the feedback, and have addressed specific points below.
>
> ##### ”Supervised Learning vs RL” #####
> In this setting, both Supervised Learning and RL learn markovian policies because there is no additional dependence on previous states. However, supervised learning is less able to associate important moves with their **time delayed** reward. Motivated by your comments, we ran another experiment to explore this (Figure 6 in the new version), where we looked at the number of “fatal mistakes” made by supervised learning vs RL: a fatal mistake being one where the agent makes an irrecoverable error. We found that supervised learning is *much* more prone to fatal mistakes, explaining the worse performance, and validating our conjecture that “reinforcement learning is learning to focus most on moves that matter for winning”
>
> ##### “ Why does multiagent generalize better than single agent defender” #####
> Training in the multiagent setting likely means the defender sees a greater diversity in the data, resulting in a more robust learned policy. Exploring this further could be interesting future work!
>
> ##### Summary #####
> Thank you for the kind comments! We also believe that it is valuable and unique contribution to have a challenging game but with linearly expressible optimal policy to study RL, make comparisons to Supervised Learning and explore Generalization.

---

> > ### Comment · AnonReviewer1 · 2018-01-10
> > **rebuttal response**
> >
> > Thank you for running the additional experiment, comparing your supervised learning setup to the RL approach.  I think this an interesting empirical distinction, that a supervised learner makes more fatal mistakes as the game becomes more complex.  But I think this raises more questions: What is the fundamental difference between these two approaches that leads to this performance gap?  Can RL be seen as a minimax routine, whereas the supervised learner may perform well on average?  Is there a way to fairly compare/construct different objectives these two approaches may be optimizing?  I believe you've highlighted an interesting phenomenon, but I wish more understanding had been cultivated by its analysis.
> >
> > I am keeping the same score; I still think this is interesting work, but I think the paper can be improved by coupling the empirical study with more analysis.

---

> ### Author Response · Authors · 2018-01-03
> **Response to Revision and Rebuttal?**
>
> Dear Reviewer,
>
> Happy new year! We would be very grateful to hear your responses to our paper revision and rebuttal. In particular, we believe that the paper revision has additional figures that answer some of your questions.
>
> Best,
>
> The Authors

---

### Official Review · AnonReviewer2 · 2017-11-29
**An interesting new RL benchmark, but too much uncertainty in the experiments**

**Rating:** 6
**Confidence:** 3

**Review:**

This paper presents an adversarial combinatorial game: Erdos-Selfridge-Spencer attacker-defender game, with the goal to use it as a benchmark for reinforcement learning. It first compares PPO, A2C, DQN on the task of defending vs. an epsilon-sub-optimal attacker, with varying levels of difficulty. Secondly it compared RL and supervised learning (as they know the optimal actiona at all times). Then it trains (RL) the attacker, and finally trains the attacker and the defender (each a separate model) jointly/concurrently.

Various points:
 - The explanation of the Erdos-Selfridge-Spencer attacker-defender game is clear.
 - As noted by the authors in section 5, with this featurization, the network only has to learn the weight "to multiply" (the multiplication is already the inner product) the feature x_i to be 2^{-(K-i)}, K is fixed for an experiment, and i is the index of the feature, thus can be matched by the index of the weight (vector or diagonal matrix). The defender network has to do this to the features of A and of B, and compare the values; the attacker (with the action space following theorem 3) has to do this for (at most) K progressive partitions. All of this leads me to think that a linear baseline is a must-have in most of the plots, not just Figure 15 in the appendix on one task, moreso as the environment (game) is new. A linear baseline also allows for easy interpretation of what is learned (is it the exact formula of phi(S)?), and can be parametrized to work with varying values of K.
 - In the experimental section, it seems (due transparent coloring in plots, that I understand to be the minimum and maximum values as said in the text in section 4.1, or is that a confidence interval or standard deviation(s)? In ny case:) that 3 random seeds are sometimes not enough to derive strong conclusions, in particular in Figure 9.
 - Everything leads me to believe that, up to 6.2, the game is only dealt with as a fixed MDP to be overfit by the model through RL:
   - there is no generalization from K=k (train) to K > k (test).
   - sections 6.2, 6.3 and the appendix are more promising but there is only one experiment with potential=1.0 (which is the most interesting operating point for multiagent training) in Figure 8, and potential=0.999 in the appendix. There is no study of the dynamics of attacks/defenses (best responses or Nash equilibrium).

Nits:
 - in Figure 8, there is no need to plot both the attacker and defender rewards.
 - Figure 3 overwrites x axis of top figures.
 - Figure 4 right y-axis should be "average rewards".

It seems the game is easy from a reinforcement learning standpoint, and this is not necessarily a bad thing, but then the experimental study should be more rigorous in term of convergences, error bars, and baselines.

---

> ### Author Response · Authors · 2017-12-17
> **Response to Review 1**
>
> Thank you for your time in reviewing the paper and your comments! We’ve uploaded a new version of the paper based on the feedback, and have addressed specific points below.
>
> ##### “A linear baseline is a must have” #####
> We’ve added a new subsection and results (Figure 2, section 4.1.1.) where we show the performance of linear models that are trained with PPO, A2C and DQN. While theoretically, a linear model is expressive enough to learn the optimal policy, in practice, we see a large improvement in using a deeper model. (We had also observed this in initial experiments with the environment, but omitted it due to the better performance with the deeper models.)
>
> ##### “Three random seeds not enough in particular in Figure 9” ######
> We reran the experiment for the paper v1 figure 9, now figure 10, with 8 random seeds. Due to the larger number of seeds, we plotted the mean and show shaded the standard deviation. The figure shows even more clearly the better generalization of multiagent training over single agent training.
>
> ##### Figure 4 (in original paper) now Figure 5 #####
> Thanks for your comments, we’ve edited Figure 4 (old version), now figure 5 to make the main message clearer: supervised learning does better on a per move basis, but does worse at playing the game.
>
> ##### Fatal Mistakes (Figure 6) #####
> We’ve also interpreted this result further, and show that this performance difference is likely due to supervised learning making many more fatal mistakes (Figure 6) -- errors in play that cannot be recovered from.
>
> #####Figure 8 (original paper) now Figure 9#####
> Thanks for the comment, we’ve removed the dashed lines to make the performance clearer.
>
> #####Multiagent training at potential 1.0#####
> There aren’t many plots in the paper with potential=1.0 for multiagent training because training an attacker agent successfully is much harder than training the defender agent (larger action space), and for larger K, the attacker performs poorly at potential~1.0, with the defender typically dominating.
>
> ##### “There is no generalization from K=k train to K > k (test)” #####
> As we understand it, this would involve picking a K_0 for training, and then testing on K > K_0 during test time. But we don't see a way for this to produce useful insights, since if the model has never seen pieces at levels other than the K_0 levels shown at train time, we cannot expect that it will learn the correct weighting for the levels it hasn’t seen at all.
> We therefore try the converse of this, where we train on K_0, and test on K,  K < K_0, and find that decreasing K does not improve play. While this does suggest that the model is overfitting to K_0, we believe this is an interesting phenomena, highlighting some of the weaknesses of the current methodology. Determining how to adapt our methods to enable this generalization across different K would be exciting to explore in the future.
>
> ####Summary####
> We believe we have addressed the main points of your response (linear baselines, more seeds, additional interpretation plots) as well as clarified certain points of confusion (multiagent training at potential 1, generalization at different levels.) Our results present a environment that has variable difficulty and is challenging to learn, but also a known, simple optimal policy to compare to. The environment demonstrates many of the typical phenomena observed with RL and provides insights into Supervised Learning vs RL, the effects of multiagent play, and also generalization and catastrophic forgetting. We strongly believe that further work on this environment will help develop more robust RL methods.

---

> > ### Comment · AnonReviewer2 · 2018-01-05
> > **Revising my 5 in 6**
> >
> > Thanks for the additional work, I think it makes the paper better, and puts data behind things that were previously claimed with too little support. I am still a bit unconvinced as far as the modeling goes, as there indeed seems to be overfitting going on, but I revise my overall 5 mark into 6, as your update renders the ESS game more interesting as a "simple" benchmark for RL algorithms (at least your update gave me a better comprehension of why ESS is interesting).

---

> ### Author Response · Authors · 2018-01-03
> **Response to Revision and Rebuttal?**
>
> Dear Reviewer,
>
> Happy new year! We would be very grateful to hear your responses to our paper revision and rebuttal.
>
> Best,
>
> The Authors

---

### Decision · Program_Chairs · 2018-01-29
**ICLR 2018 Conference Acceptance Decision**

**Decision:**

Invite to Workshop Track

**Comment:**

The paper introduces an interesting family of two-player zero-sum games with tunable complexity, called Erdos-Selfridge-Spencer games, as a new domain for RL.  The authors report on extensive empirical results using a wide variety of training methods, including supervised learning and several flavors of RL (PPO, A2C, DQN) as well as single-agent vs. multi-agent training.  The reviewers also appear to agree that the method appears to be technically correct, clearly written, and easy to read.

A drawback of the paper is that it does not make a *significant* contribution to the field.  In combing through the reviewer comments, none of them identify a significant contribution.  Even in the text of the paper, the authors do not anywhere claim to have made a significant contribution. As the paper is still interesting, the committee would like to recommend this for the workshop track.

Pros:
        Interesting domain with tunable complexity
        High-quality extensive empirical results
        Writing is clear

Cons:
        Lacks a significant contribution
        Appears to overlook self-play, the dominant RL training paradigm for decades (multiagent training appears to be related but different)
        Per Reviewer3, "I remain unconvinced that these games are good general tests for Deep RL"